# Wading through Molasses: A qualitative examination of the experiences, perceptions, attitudes, and knowledge of Australian medical practitioners regarding medical billing

Margaret Faux [1,2]*, Jon Adams[1], Simran Dahiya[3], Jon Wardle[1,4]

1 School of Public Health, Faculty of Health, University of Technology, Sydney, Australia, 2 Solicitor of the Supreme Court of NSW and the High Court of Australia, Parkes, Australia, 3 Faculty of Medicine, University of New South Wales, Kensington, Australia, 4 Faculty of Health, Southern Cross University, Lismore, Australia

* margaret.a.faux@student.uts.edu.au

**Data Availability Statement:** The data for this study consists of transcripts of 27 participants that contain identifying information. The data cannot be

## Abstract

### Background

Medical billing errors and fraud have been described as one of the last "great unreduced healthcare costs," with some commentators suggesting measurable average losses from this phenomenon are 7% of total health expenditure. In Australia, it has been estimated that leakage from Medicare caused by non-compliant medical billing may be 10–15% of the scheme's total cost. Despite a growing body of international research, mostly from the U.S, suggesting that rather than deliberately abusing the health financing systems they operate within, medical practitioners may be struggling to understand complex and highly interpretive medical billing rules, there is a lack of research in this area in Australia. The aim of this study was to address this research gap by examining the experiences of medical practitioners through the first qualitative study undertaken in Australia, which may have relevance in multiple jurisdictions.

### Method

This study interviewed 27 specialist and general medical practitioners who claim Medicare reimbursements in their daily practice. Interviews were recorded, transcribed, and analysed using thematic analysis.

### Results

The qualitative data revealed five themes including inadequate induction, poor legal literacy, absence of reliable advice and support, fear and deference, and unmet opportunities for improvement.

shared publicly due concerns of participant confidentiality and ethics requirements. Interviews were confidential to enable freedom of expression by participants, and participants consented to the study with the understanding that only de-identified quotations would be made public, not the entirety of the transcripts. Therefore, only illustrative quotes from the transcripts have been included in this paper. Data access requests may be made to the University of Technology Sydney at Research. ethics@uts.edu.au.

**Funding:** The authors received no specific funding for this work.

**Competing interests:** I have read the journal's policy and the authors of this manuscript have the following competing interests - Margaret Faux is the CEO of a medical billing company.

## Conclusion

The qualitative data presented in this study suggest Australian medical practitioners are ill-equipped to manage their Medicare compliance obligations, have low levels of legal literacy and desire education, clarity and certainty around complex billing standards and rules. Non-compliant medical billing under Australia's Medicare scheme is a nuanced phenomenon that may be far more complex than previously thought and learnings from this study may offer important insights for other countries seeking solutions to the phenomenon of health system leakage. Strategies to address the barriers and deficiencies identified by participants in this study will require a multi-pronged approach. The data suggest that the current punitive system of ensuring compliance by Australian medical practitioners is not fit for purpose.

## Introduction

Medical billing errors and fraud have been described as one of the last "great unreduced healthcare costs," with some commentators suggesting measurable average losses from this phenomenon are 7% of total health expenditure [1]. It is therefore central to the long-term economic viability of any health system that medical practitioners have clarity and certainty around relevant billing standards and rules. However, a growing body of international research, mostly from the U.S, suggests medical practitioners are ill equipped to understand the complexities of the health systems in which they work.

Like the reported experiences of their U.S colleagues, evidence suggest Australian medical practitioners may be experiencing difficulty navigating complex medical billing rules [2]. It has been suggested that the rate of non-compliant billing under Australia's Medicare caused by deliberate abuses by medical practitioners is between 10–15% [3]. However, how much non-compliant billing is deliberate is uncertain, as it rests in a spectrum with criminal fraud at one end and unintentional errors at the other and currently the precise quantum of each is unknown. This is largely because the problem is not what can be seen, but what cannot. Lax regulation, poor administration, system complexity and the fact that medical practitioners are never taught how to use the system correctly at any point in their careers have all been cited as factors contributing to this problem [4]. Increasing complexity has occurred in tangent with increased penalties for non-compliance [5] and pressure on medical practitioners to bill correctly has reached the point where some authors have suggested that compliance with Medicare billing rules has become a contributing factor to medical practitioner burnout and suicide [6]. However, one area of activity that has been overlooked is improving user knowledge of the medical billing system.

Multiple recent U.S studies on the topic of medical billing literacy [7] have consistently reported demonstrably low literacy which may be improved by targeted educational initiatives, including by medical billing and coding education being a mandatory inclusion in the medical curriculum. However, an apparent inertia to act persists. In Australia, discussion around this topic is less mature, with very little similar research having been undertaken.

The aim of this study was therefore to address this research gap by examining the experiences of Australian medical practitioners in grass roots practice as they interact with Medicare and claim reimbursements under Australia's unique Medicare Benefits Schedule (MBS) codes [8]. This study will also explore medical practitioner knowledge of medical billing requirements, attitudes and perceptions to Medicare, and seek to identify any barriers to compliance as well as exploring possible solutions to deficiencies in current arrangements.

## Methods

Between July 2016 and May 2019, semi structured interviews were conducted with specialist and general medical practitioners both of whom are required to claim Medicare reimbursements in their daily work. The study was geographically restricted to the State of New South Wales, was approved by the relevant Human Research Ethics Committee and consent was obtained from all participants. Participant information has been de-identified to preserve anonymity.

### Participants

Twenty-seven interviews were conducted, twelve with General Practitioners (GP) and fifteen with Salaried Medical Officers (SMO), the latter of whom are specialists working in Australian public hospitals. Participants were recruited through advertising with their professional associations, direct approaches and "snowballing". Participant demographics included 11 females and 16 males and a mix of overseas and Australian trained medical practitioners, who worked in both regional and city locations. The full spectrum of career stages was represented, including early career stage medical practitioners (defined as 0–7 years post-graduation) through to those who had practiced medicine for over 30 years. The SMO cohort included a variety of procedural and non-procedural specialists.

### Data collection

Medical practitioners who responded to initial contact were sent an information sheet (S1 File), consent form (S2 File) and a short overview of the research via email, and those who participated signed the consent form prior to the interview.

Although every effort was made to identify participants who were not known to the principal researcher (first author), being someone who has worked in the medical billing industry for over 30 years it was likely that some participants would have a coexisting relationship. One GP and one SMO were personally known to the principal researcher, and another GP and SMO were professionally known. In addition, three SMOs were professional acquaintances. While this was unavoidable, it is not uncommon in qualitative research projects (for example a nurse questioning other nurses in their organisation as part of a project).

To ensure personal relationships (none of which were close) did not cloud data collection, the principal researcher continued to have regular discussions with other members of the research team adopting reflective practice to eliminate bias and ensure research integrity. Further, the third author listened to the audio recordings of all interviews and provided important insights when reviewing the draft paper to ensure data were accurately reflected and reported, with additional input from other authors as required.

To address possible conscious or unconscious bias, triangulation was used where an experienced qualitative researcher separately analysed and interpreted the data and any differences in researcher perspectives were cross checked to arrive at an overall interpretation. By implementing these accepted methods rigour, trustworthiness, authenticity and credibility were addressed [9].

As this study forms part of the doctoral thesis of the principal researcher, it was incumbent upon her to personally conduct as much of the work as possible. However, this project was at all times closely supervised by the last author, who is a senior researcher experienced in qualitative data collection. The principal researcher had ongoing discussions with the last author throughout the data collection phase and during the analysis and coding of the data.

Further, to ensure research integrity the last author directly sat in and supervised the first two interviews (including with the GP who had a personal relationship). Following approval of

the first two interviews, the principal researcher continued and personally conducted all 27 interviews. Most of the interviews were conducted in person (n = 23) at a place and time convenient to the participants. Due to geographical barriers, some of the regional GP interviews were conducted by phone (n = 4).

Two listeners and two independent coders analysed the data in line with qualitative research norms. The third author listened to the audio recordings of all interviews and edited final transcripts to ensure accuracy. After discussion with the last author regarding emergent themes, the first and third authors worked together to code the data, with the other authors reviewing in areas that required resolution to disagreements.

The interviews were semi-structured, with a question sheet used to loosely guide questioning. A copy of the question guide is shown as S3 File. Participants were encouraged to speak freely and openly and were given unlimited time to enable full exploration of the topic. The interviews continued until theme saturation had been reached, the average interview length was one hour, and all participants consented to the interviews being recorded. The interviews were subsequently transcribed.

## Data analysis

The process of data analysis included the five documented steps using the framework approach which is broadly described as familiarisation, identification of framework, charting, mapping and interpretation [10].

The principal researcher reviewed the manuscripts to familiarise herself with the data including reading and re-reading the transcripts, relistening to the audio files, organising the data for analysis, visually scanning the transcripts and beginning the process of sorting the data to consider its overall meaning. Identification of the framework was then undertaken to draw out key themes and issues from the text around which the data were then organised. The data were then indexed to identify themes and finally, mapping and interpretation was undertaken, whereby associations were clarified, and explanations worked towards.

In order to ensure quality during data analysis, quality assurance measures based upon systematic and self-conscious practice were implemented [9]. A self-reflective, critical examination of potential bias was also undertaken by the principal researcher, who spent prolonged time in the field engaging with the subject matter.

## Findings

Analysis of the qualitative data revealed five themes related to Medicare and MBS billing, including inadequate induction, poor legal literacy, absence of reliable advice and support, fear and deference, and unmet opportunities for improvement. Examples of raw data analysis and themes are shown in Table 1.

### Inadequate induction into Medicare and MBS billing

All participants reported their first experience generating a medical bill, or claiming to Medicare, taking place in a knowledge vacuum, where they felt inadequately prepared. As the following quotes suggest, many respondents reported little–if any–training, and if training did occur it was usually brief, informal and taught by someone who may not necessarily have been qualified to teach it:

"...when I did my GP training we had a block of training prior to our very first day on the job...we basically just learnt you know your 23 and 36 item number [common time based

**Table 1. Example of raw data analysis.**

| Raw Data | Theme |
|---|---|
| [Interviewer asked SMO7 if education at various levels adequately equipped him to bill correctly] Not at all. It is purely through by necessity to understand it oneself and to understand the vagaries not only of billing, but how it works in the context of the staff specialist or ward arrangements, which are quite complex. [interviewer: 'any education on that either?'] No zero. Zip. | Inadequate induction |
| Bulk billing, I understand is where whatever Medicare says, so if . . . I treat the patient for say keeping on breathing machine let us say. Government says you can earn $50 a day for doing that and bulk bill would be if I say okay give me $50. If I charge $60, then I have charged a gap. [interviewer: when can you do that? SMO12 replied] No idea. | Poor legal literacy |
| [interviewer]. . .so when it goes off into accounts, how confident are you about what happens next? [SMO14] I am confident because as the director, I have explored that, my colleagues would be somewhat less confident. [interviewer] With item numbers. . .? [SMO14] No just total numbers. Just money. Could have been anything. So, in fact, in reality I have no idea. [interviewer] So. . .you have got an idea of the total dollar amount that is billed, do you have an idea of the actual item numbers? [SMO14] No, not at all, not a jot, not one single solitary scintilla. | Absence of reliable advice and support |
| [GP3] We have a practice manager and we have asked her to contact Medicare about some. . .uncertain issues regarding Medicare. . .and she will get five different answers from five different people that she rings. . .that is a regular experience and I say ". . .there's no point in ringing Medicare about this" because I do not know who she is speaking to. I do not know whether she is speaking to a manager. . .or somebody who has recently started in Medicare who does not have much experience. . .and is just reading from one part of the manual but doesn't know the other parts. . .we've always had that experience if you ring up. . .the most recent example. . .charging through Medicare for overseas travel. . .she has spoken to several different people and received different answers from each one. | |
| [GP2] I probably underbill. . .I'm just going to do what I know is safe. | Fear and Deference |
| [GP4] The threat of audit kind of hangs over. . . | |
| [SMO7] I do not order a lot of blood tests. I do not order a lot of scans. I am very interested in. . .evidence base, I am interested in doing what is needed, I try not to pander to anxiety, it's very difficult, it is much easier to give in and just order a million tests. . .It is an impost on the national health, so I think there is a responsibility. | |
| [GP8] Sending some more resources . . .for educating the doctors, by various means be it sending them letters like case examples, emails, having some conferences around, you know, correct Medicare billing etc and educating doctors the implications of incorrect charging particularly over-servicing and fraud, I think that is very important. Doctors just learn from their colleagues and others, you know, we are hearing stories, it is not something they are actively involved in, so there should be an education process and may be even attaching some category points. . .if the doctors understood Medicare and I think that is very important. The system is there but is not enough education about it. | Unmet opportunities for improvement |

attendances] [11]. . . there would have been question and answer time, but we hadn't prac-tised yet so we wouldn't really have known what questions to ask." (GP1)

". . .in that induction program there was a guide to claiming, a very brief guide. I think my experience and a lot of other GP trainee's experience was that we had no idea, we were out there, kind of at the coal face, I had zero idea of what we were doing and. . .it was like walking through molasses, it was very hard to negotiate. . .It is so hard to understand, ridiculous. . ."(GP4)

"[I was] totally naïve, I just believed what he said, thinking he is my senior guy and that was it, so I had no idea that there are legal implications, I had no idea." (GP7)

While most GPs reported a brief induction process, SMOs reported having no induction at all, as explained by the following SMOs:

*"Um trial and error, there was no formal introduction, no formal training as you go through. . . there was no mention of billing. . .so you navigate it by the skin of your teeth."* (SMO11)

*"I had no idea how Medicare kind of worked . . .no one taught me how to bill. . .I had no idea what it meant to Medicare bill, what gaps were, what scheduled fee was, all the different rates of things were, so it made no sense. . .there is absolutely no training."* (SMO1)

*". . .when you are a Registrar and when you finish you then realise, oh, there is Medicare. Now what have I been taught about Medicare? Essentially nothing. . .you realise you are supposed to bill, but still have no inkling how to do it."* (SMO10)

## Poor legal literacy of Medicare and MBS billing

When participants were asked detailed questions about fundamental legal requirements to bill correctly, their levels of literacy were variable and some were confused in important areas, such as when it is permissible to charge a gap and what bulk billing was. Bulk billing is a common term in Australia, describing a transaction for a medical service wherein the patient does not pay any money because the medical practitioner chooses to accept the amount of the available government subsidy for that service [12]. The term 'gap' in the Australian context refers to a patient out-of-pocket payment which in many countries is described as a co-payment.

Both of the following quotes were from bulk billing doctors, one of whom did not know the process he was using was bulk billing and the other was unaware he could charge a gap if he wanted to.

*". . .bulk billing, we do not do bulk billing. . .really my understanding is it is something that happens in general practice. . ."* (SMO9)

*"I think a gap would only be payable if the patient is in hospital where. . .they have to pay the gap between the doctor's fee and the health fund rebate or gap between the specialist fee and the Medicare rebate, I am not entirely sure of this; I am just guessing from the limited amount of information that I have."* (GP8)

When SMOs were asked their understanding of relevant law around bulk billing or charging gaps to patients in public hospital outpatient departments many of their responses highlighted a deep lack of knowledge.

*"I think if we as the department decided to charge a gap, we can . . .there might be a specific rule, like you cannot charge a gap, but I am not sure, I have never asked questions, I have wondered about it though."* (SMO3)

*"Can a gap be charged? I actually do not know the answer to that question."* (SMO4)

*"[billing in the public hospital is] a minefield. My understanding is that for outpatient services in a privatised clinic like this it's quite within our rights to charge a gap,"* though when quizzed about the source of that information he said, *"Look I do not know the precise details of that; this is just something I have been told."* (SMO6)

Confusion about the legalities of this area of public hospital billing extended to GPs, with one GP incorrectly asserting that bulk billing in public hospital outpatient departments is illegal.

> *"the states are fraudulently thriving on Medicare, in all public hospitals. . .the practice is frightening. . .they bulk bill you in the public hospital* [outpatient department]*."* (GP5)

The majority of participants were also unclear about fundamental billing requirements. In Australia's gatekeeper model health system, patients usually require a valid referral from a general practitioner before seeking more specialised care. However, most participants did not know what constituted a valid referral. Other very basic requirements to bill correctly were also poorly understood by most participants such as specific rules around billing eligible war veterans, and whether any patient has to sign a form when the medical practitioner bulk bills the patient.

> *"Valid referrals, I do not know, I have no understanding of that. I am actually unsure."* (GP9)

> *". . .there seems to be at least as far as I am aware (but no one really knows) a practice that anyone who holds the Veterans Affairs Card will not be charged a gap. Whether that is true or not, I do not know."* (SMO4)

> *"I am not really sure, to be honest. . .I am not sure if it is compulsory,* [the bulk bill form] *needs to be signed by the patient. I do not really know."* (GP9)

When participants were asked how well they thought they complied with current standards some did not know what the standards were or whether such standards existed, and very few participants were aware of the penalties for noncompliance.

> *"I actually don't know that we would meet the criteria because I don't really know what they are."* (SMO15)

> *"I don't really know. . .I mean I am sure they could make you pay back the money and there probably is jail time eventually at some point, but to be honest I don't really know what the penalties are."* (GP1)

## Absence of reliable advice and support

The majority of participants tended to describe their experiences seeking support and advice from Medicare in negative terms and preferred to direct medical billing questions to practice managers, colleagues, hospital finance departments, professional organisations and in one case, social media.

> *". . .there was something recently that we actually called them up for and then it was some huge kerfuffle and. . .it kept going round and round. . ..it was about this item number and they just kept reading the same thing we were reading, which was ambiguous. So, it was an utter waste of time."* (GP12)

> *"I always felt like the advice was pretty good but if it got too technical, they were fudging it."* (SMO15)

> *"We get three different answers literally, about the same thing."* (GP5)

When asked what gave participants confidence in the medical billing expertise of others, their responses expressed blind faith, difficulties obtaining reliable advice and support and the need to trust someone, as the following quotes demonstrate.

*". . .the assumption is that. . .the secretarial staff would have done that before and they will be doing it for other doctors but whether they have had specific training in the rules and regulations around Medicare etc one never really knows. . .whether they had original training in what was actually required and what was not etc, I suppose it is not something that is very well regulated."* (SMO4)

*"Looks and appearance, she* [the Practice Manager] *just appeared to know what she was doing, and I trusted her. . .I had to."* (GP6)

*"the bottom line is it* [MBS billing] *is not clear, and it is not easy to get clarity about some of those issues."* (GP3)

A private Facebook group had become the main source of Medicare billing information for one GP, who felt it was authentic and relying on it would protect her in the event of an audit.

*"I do not have a choice but to rely on that because I do not think there is anything else and I realise the problem. If there are other things available, they're not made obvious to us, and I am someone actively seeking out this information. So, if I am looking for it and this is the best that I can find, what would a reasonable group of my peers do differently to what I am doing? Could I rely on that to be investigated? I have to, and I think that that is all I can do because I do not think there are other options. . ."* (GP4)

SMOs reported a preference to seek support from inside the hospitals where they worked, even though some said they didn't know who to ask and others described the information they received as inherently unreliable. No SMO mentioned referencing the National Health Reform Agreement (NHRA) [13], which is the key agreement between the State and Federal Governments containing the rules for medical billing in public hospitals.

*"I just feel dumb at these things, I need someone to explain it really in very basic terms to me. The area of private practice billing* [in public hospitals] *really baffles me."* (SMO3)

*"I knew nothing* [about billing in public hospitals] *so they* [the hospital finance department] *had to know more than nothing,"* (SMO7)

All but one participant described education on medical billing throughout their careers in clear, unambiguous terms, summarised by the following typical response.

*"[it was] absolutely, totally, totally [inadequate]. Part of the problem, it is very interpretation based, there is no clarity on it. That's really poor and there isn't, to my knowledge, any kind of place that we can go, that in a succinct fashion, in a way that we need it to be, we can have very clear guidance about what we can or we cannot do and I strongly feel that I've had to wing this in terms of pulling stuff together, to make my own knowledge on it."* (GP4)

Most participants understood they were personally responsible for billing, but all had arrangements in place whereby third parties administered billing on their behalf. The advantage of this arrangement was reported as saving time, and the disadvantage was the inherent risk in having diminished control and visibility over the final item numbers submitted to

Medicare. SMOs in particular were not confident that the item numbers they put on hospital forms were the same item numbers that were sent to Medicare, because they had very little control over medical billing activities undertaken in their name by the public hospitals where they work.

> *". . .billing under my name in the public hospital in the outpatient department. . .I cannot see. I could not tell you if anyone did it fraudulently or inappropriately."* (SMO7)

> *"As far as the data entry from my perspective, I know that the Medicare billing is correct because I put it in, so the question is two-pronged because one is my part of it and the second part is the part that I do not do. . .there is a gap there, so I do not know about the second part, because I have not checked."* (SMO2)

> *". . .I trust my colleagues but at the end of the day I have no idea."* (SMO11)

> *"I have no control over claiming so I feel very uneasy with the whole process."* (SMO10)

Many GPs also expressed concern that they ultimately did not know or have any visibility or control over what was being submitted to Medicare in their names.

> *". . . I actually have no idea that they do what I ask them to do. I have to trust them, which I do of course. But they could be submitting all sorts of weird and wonderful things and I confess that I don't know what they're doing. . .you have got to trust someone."* (GP3)

> *"There's that element of, I'm legally responsible for it and yet someone else is actually pressing the buttons, and maybe there is room for error there that I'm actually liable for, which I haven't even thought about, which is a bit disturbing."* (GP2)

All participants described the unreliability of medical billing advice no matter who provided it, but perhaps the most startling example describing the unreliability of government advice was from a SMO who had been audited. This participant described her correct application of a locum billing rule, whereby when acting as a locum for a colleague, the medical practitioner is not permitted to claim an initial attendance item, but must instead claim a subsequent attendance item when a colleague has already reviewed the patient. The participant was subjected to what appears to have been a mishandled audit by Medicare, who appeared to have misunderstand the operation of the rule, which at all relevant times was clearly described in the MBS. As a result of the audit and Medicare's failure to explain to the SMO what she did wrong (which may have been nothing), the SMO changed her billing behaviour and is now billing incorrectly and costing Australian taxpayers more.

> *"I got audited. . . I then rang Medicare back and I said, "this was the logic for why I claimed 116* [a subsequent consultation]*" and I said, "Is this correct or not correct?" And they said, "we are not supposed to advise on the phone." And then I said, "So for me to get some advice, where can I go?" And they said, "you have to look at the MBS schedule." And I said, "I looked at the MBS schedule, I can't find the answers and I have asked my colleagues what they do and half of them do what I do and half of them put 110* [an initial consultation]*." So, I never got the right answer. They said they cannot provide any answers. It's pretty poor. I think there are answers that sometimes, you know, you're not quite sure, but don't really know who to ask except for your colleagues and sometimes I feel like the colleagues probably just make it up anyway because they probably don't know.* [after the audit] *I did change my practice and now I use a 110 when I'm covering somebody else"* (SMO10)

## Fear and deference

Most participants spoke positively about Medicare as a health system, describing its purpose as being to provide universal health coverage irrespective of ability to pay, and acknowledged the nexus between their billing and their responsibility for the national health budget. However, some participants commented on the shortcomings and inherent vulnerabilities in an honour-based scheme such as Medicare.

> *"I think we are the gatekeepers of it really, and the responsibility is on us as the doctors who are claiming. I think we need to be really quite careful about how we claim because I think if we are not claiming appropriately, then our health budget is not going to be able to sustain, you know, future healthcare."* (GP9)

> *". . .you have rights to minimise cost to a country and then you have the rights to the patient in front of you, and sometimes that doesn't marry." (*GP12)

> *"Well, the opportunity for cheating is as you can imagine endless. The way you describe your service is entirely up to you. . .I think most people are not dishonest and most doctors are not dishonest, but still as a taxpayer I do not like a system where you can endlessly plunder the public purse with relatively blunt scrutiny." (*GP10)

Most participants described billing defensively on occasions due to fear and anxiety of Medicare audits. One participant said she was initially scared of Medicare and recalled thinking when she first started practice, *"I will just stick to my 23s* [11] *and then I won't do anything wrong."* (GP1)

Under-billing was commonly reported, with many participants saying they would always contact Medicare to refund payments if they had made an over-billing mistake but would not correct under-billing errors. One respondent gave a typical response on this issue, *"If there is any doubt, I just do not claim it, it is as simple as that. I have a career of more than 20 years and I don't intend to end it prematurely."* (GP5)

Most participants also said they were not comfortable talking about money with their patients, so preferred to have the money handled by someone else and the majority expressed a disinterest in billing, with one respondent providing a typical response, *"I think no doctor wants to do their billing themselves, if I have to do billing myself, I probably would not do this."* (GP5)

## Unmet opportunities for improvement

A prominent theme was a desire for the current educational deficit to be addressed. Participants had mixed views about the precise place and format of medical billing education with some suggesting a blended approach, whereby content would be provided both at the undergraduate level, and technical details taught later as required.

> *"I think if doctors in training have a very good understanding of how hospitals run, how Medicare works, how a private practice works, they will from the very beginning be much more engaged in trying to ensure that the funding is provided in an equitable manner and it is not trying to rort the system or do anything like that but is being aware of how things work. . .I think it is essential."* (SMO4)

> "[The educational deficit is a] *massive gap. . .if people are going to be working in the Australian Health System, they need to understand the remuneration and how it occurs in our health*

*system, I think health economics is equally important and there is nothing taught about health economics."* (GP7)

*"A lot of people would look at medicine and say, well look, people seem to get good salaries and a good lifestyle and that sort of thing. . .to understand that isn't just going, "so well, doctors seem to be having a good time, but I don't really want to know the mechanism of it." I think understanding the mechanism is really important."* (SMO1)

A common view about the practicalities of any future medical billing education suggested an applied learning approach would be more helpful than expecting medical practitioners to understand and interpret *"legal wording."* (GP8)

## Discussion

### General knowledge of medical billing and the impact of third parties

The qualitative data presented in this study suggest Australian medical practitioners are ill-equipped to manage their Medicare compliance obligations, have low levels of legal literacy and desire education, clarity and certainty around complex billing standards and rules. This is consistent with the results of prior survey findings in Australia [4] as well as findings in other countries such as the U.S and Canada [14–16]. This finding also aligns analysis of Australian medical billing policies which reported that a single Medicare service in Australia can be the subject of more than 30 different payment rates, multiple claiming methods and myriad rules [17].

The data also suggest the current 'rules' of medical billing are confusing, and medical practitioners are struggling to understand and apply them in daily practice.

All participants commented on the potential negative impact of untrained third parties administering medical billing on their behalf. Participants described this common operating model as reducing the practical control and visibility they had over bills submitted to Medicare in their names, and was an area in which the law was out of step with the realities of modern medical practice management.

### Risks to State and Federal Government relations and public hospital funding

Responses from participants suggested that while most medical practitioners have an awareness of the existence of the MBS (though many did not access or use it), they had no knowledge of the vast interconnected body of law that impacts their daily billing decisions, most notably the NHRA [13]. The apparent lack of awareness of the NHRA by SMOs combined with demonstrably poor understanding of some of the most basic elements of correct billing such as the components of a valid referral, may have serious repercussions extending beyond individual practitioners. Whilst SMOs are required to comply with the complex provisions of the NHRA, they are not parties to it, so cannot personally breach an agreement they did not sign. The relevant signatories to the NHRA are the Federal and State Governments, the latter of whom may be exposed to investigation and substantial repayments to the Commonwealth caused by incorrect billing by the SMOs in their employ. This risk was recently identified by both the Victorian Auditor General [18] and the Independent Commissioner Against Corruption in South Australia [19], and was illuminated in this study.

This studies' data suggest SMOs may be unaware of the components of a valid referral despite this being a central component of a correct bill in a public hospital outpatient department. This finding, coupled with opaque legal drafting, inconsistent law making as between

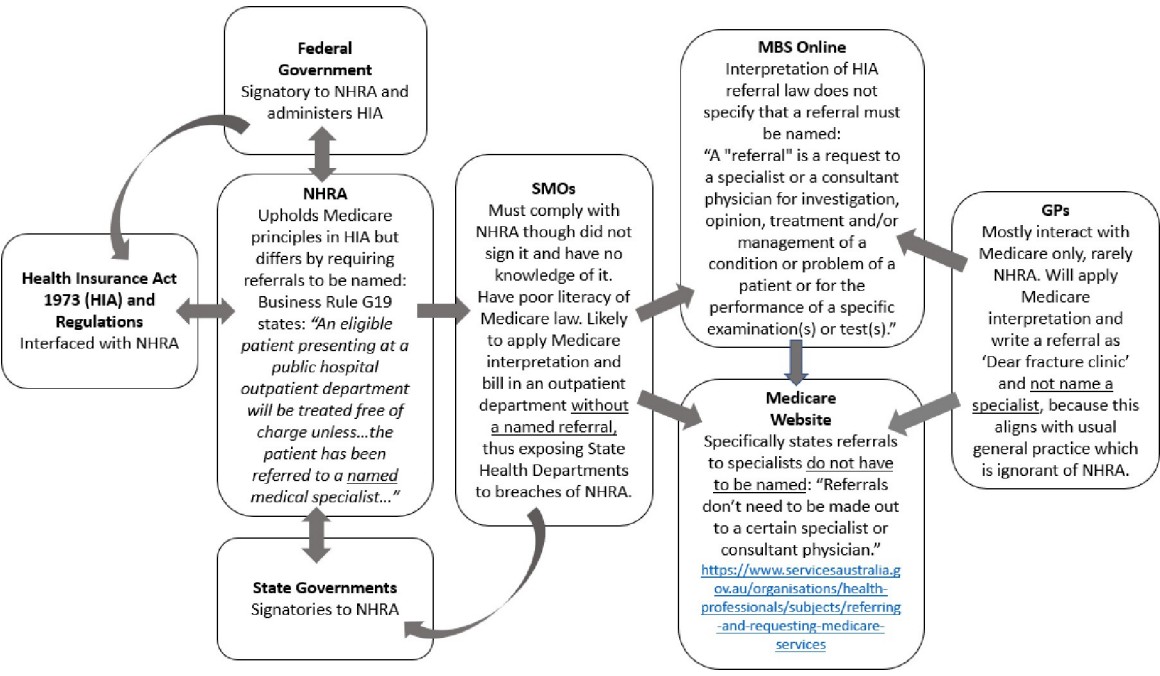

**Fig 1. Referral law inconsistencies between Medicare and NHRA and potential impact.**

the NHRA and the *Health Insurance Act 1973 (Cwth)*, (which has been the subject of earlier critical analysis) [2] as well as inconsistent departmental interpretation of relevant legal provisions, may have extinguished any possibility of compliant billing in this important area. crippling the Federal Governments' ability to prosecute breaches when they occur. The mechanism of this process is shown in Fig 1.

## Medicare audit anxiety and cognitive dissonance

Fear of Medicare audits was another issue highlighted by some participants, which appears to be contributing to overall feelings of anxiety and unease. This has the potential to impact patient care if medical practitioners make conservative treatment choices fuelled by fear of investigation, a potential sequela that has also been reported in the US [20].

When asked about the connection between their billing patterns and their responsibility for the national health budget, participants acknowledged their responsibility to bill correctly and distribute finite resources prudently. However, this sat at odds with earlier responses around a preference by all participants to remain disconnected from billing administration, which they felt was not what they had studied medicine to do. This represented a striking cognitive dissonance in which the space between thought and action was occupied by ignorance from inadequate education, and indifference to having oversight of their own health budget spend.

## Inadequate government support

This study found no evidence of the availability of reliable advice and support for billing questions, including from Medicare, with the main sources of information being medical colleagues and administrative staff who themselves have never been formally taught how to bill correctly, but whom medical practitioners feel they have no option but to trust. Participants reported that the "the blind leading the blind" method by which medical billing information is

disseminated may be perpetuating errors and myths. Further, the consistency in the experiences of the wide cross section of participants in this study supports a finding that extremely low levels of legal literacy in relation to medical billing may be creating a vortex of misinformation contributing to health system leakage.

Further, the data suggest that a lack of administrative resources and support provided by the Australian Government may have left medical practitioners with no place to go for legally accurate, reliable advice, meaning that despite due diligence, a medical practitioner may still fall foul of the law. In one case, a participant who described correct billing practices, appears to have been led into incorrect billing by the Australian Government who may not have the appropriate resources to provide accurate interpretations of its own rules to practitioners.

The participants of this study were clear that expecting medical practitioners to comply with complex and mercurial billing laws without relevant skills or training was unrealistic. Moreover, it is suggested that denying medical practitioners access to clear, reliable advice and training prior to imposing sometimes very serious sanctions is indefensible and may be inconsistent with common law principles of natural justice [21].

## Strengths and limitations

Strengths of the study include the wide cross section of participants, information gathering in a non-punitive setting, and the diverse practice settings of participants including primary care and tertiary hospital-based care. The study also provides valuable insights into barriers to medical billing compliance and offers possible solutions for reform.

However, the qualitative data is contextually limited by the Australian context of a predominantly fee-for-service payment structure so the findings may not be generalisable, though the results are broadly comparable and consistent with reports of the same phenomenon in both the U.S and Canada [14–16]. Another limitation is the potential impact of selection bias caused by the recruitment methods wherein a participant with high ethical standards was likely to work in a practice with others having the same standards. However, any impact would have been limited to the three GP practices where more than one GP was interviewed and possibly in the public hospitals where multiple SMOs were interviewed. However, any impact is likely minimal as all participants worked and billed independently day to day, and most did not know each other. Seven of the participants were known to the principal researcher either directly or indirectly, however, any impact is also likely minimal because the line of questioning was consistent across all participants and results were cross checked multiple times by multiple researchers using the recognised methods already discussed.

## Conclusion

Non-compliant medical billing under Australia's Medicare scheme is a nuanced phenomenon that may be far more complex than previously thought. Therefore, many of the current punitive, post payment audit initiatives of the government are unlikely to succeed.

Strategies to address the barriers and deficiencies identified by participants in this study will require a multi-pronged approach which may include the development of clear, legally binding medical billing rules, nationally consistent, accurate and accessible education, and structural reform to tighten and align the underlying regulatory framework.

This is the first Australian study to examine the lived experiences of Australian medical practitioners interacting with Medicare and medical billing. Some of the experiences are shared with international experiences, and may therefore offer learnings for other countries implementing universal health coverage systems, in which payment integrity and control of

system leakage are of critical importance. The data suggest that the current system of ensuring compliance by medical practitioners in Australia is not fit for purpose.

## Supporting information

**S1 File. Participant information sheet.**
(PDF)

**S2 File. Participant consent form.**
(PDF)

**S3 File. Qualitative interview question guide.**
(PDF)

## Acknowledgments

The authors wish to thank the medical practitioners who shared their experiences with medical billing and Medicare.

## Author Contributions

**Conceptualization:** Margaret Faux, Jon Wardle.

**Data curation:** Margaret Faux, Simran Dahiya, Jon Wardle.

**Formal analysis:** Margaret Faux, Simran Dahiya, Jon Wardle.

**Investigation:** Margaret Faux.

**Methodology:** Margaret Faux, Jon Wardle.

**Supervision:** Jon Adams, Jon Wardle.

**Writing – original draft:** Margaret Faux.

**Writing – review & editing:** Margaret Faux, Jon Adams, Simran Dahiya, Jon Wardle.

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
