## [Decision Letter · Decision Letter 0]

1 Oct 2020

PONE-D-20-16346

Wading through Molasses: A qualitative examination of the experiences, perceptions, attitudes, and knowledge of Australian medical practitioners regarding medical billing

PLOS ONE

Dear Dr. Faux,

Thank you for submitting your manuscript to PLOS ONE. After careful consideration, we have decided that your manuscript does not meet our criteria for publication and must therefore be rejected.

Specifically:

The manuscript has been assessed by two reviewers, their comments are appended below.

The reviewers have raised major concerns about the study methodology, particularly regarding the selected sample and qualitative them analysis. In addition, they request further a careful consideration of the discussion to ensure that the relevance of this study is further contextualised.

I am sorry that we cannot be more positive on this occasion, but if you are able to address the reviewers comments, we would be willing to consider a resubmission for your work.

The revised manuscript should be submitted as a new submission to PLOS ONE and accompanied by a covering letter that refers to the original submission and outlines details of how the revisions have been completed.

Yours sincerely,

Sara Fuentes Perez, PhD

Staff Editor

PLOS ONE

Reviewers' comments:

Reviewer's Responses to Questions

**Comments to the Author**

1. Is the manuscript technically sound, and do the data support the conclusions?

Reviewer #1: Partly

Reviewer #2: Yes

2. Has the statistical analysis been performed appropriately and rigorously? 

Reviewer #1: N/A

Reviewer #2: N/A

3. Have the authors made all data underlying the findings in their manuscript fully available?

Reviewer #1: Yes

Reviewer #2: Yes

4. Is the manuscript presented in an intelligible fashion and written in standard English?

Reviewer #1: Yes

Reviewer #2: Yes

5. Review Comments to the Author

Reviewer #1: I like the topic and the conclusions. In terms of methodology, I have concerns about the use of practitioners (26%) with whom you were familiar. Also, you were the only researcher. Having two researchers looking for themes and reaching agreement is a stronger methodology. Medical schools by and large have no course work for clinicians in the healthcare systems within which they practice. I appreciate that you are calling attention to the problem, and presenting the experiences of the clinicians.

Reviewer #2: The topic of administrative costs of health systems is of considerable interest. Australia is an important use case to understand health care systems from a comparative basis since the Medicare system is a national health system with interesting public and private components. This paper is a survey of practitioners in Australia inquiring about their knowledge of the billing process. It is well written, and of interest to people working in this field of administrative costs. However, despite its strengths, the manuscript is likely of limited interest to a broader medical audience.

First, there is little explanation of the context of the Australia system for readers to understand Medicare and private. Second, there is a significant amount of jargon here that is unique to the Australia context (gap, etc). Finally, it is clear that the findings are an important issue for policy makers in Australia to consider, but its not clear that there are generalizable lessons for other markets.

6. PLOS authors have the option to publish the peer review history of their article (what does this mean?). If published, this will include your full peer review and any attached files.

Reviewer #1: No

Reviewer #2: No

- - - - -

---

## [Author Response · Author response to Decision Letter 0]

1 Dec 2020

REVIEWER 1:

In terms of methodology, I have concerns about the use of practitioners (26%) with whom you were familiar. 

We thank the reviewer for their comments and understand this concern. However, while this was unavoidable, it is not uncommon in qualitative research projects (for example a nurse questioning other nurses in their organisation as part of a project). To ensure any personal relationships (none of which were close) did not cloud data collection, the first author continued to have regular discussions with other members of the research team adopting reflective practice to eliminate bias and ensure research integrity. Further, the third author listened to the audio recordings of all interviews and provided important insights when reviewing the draft paper to ensure data were accurately reflected and reported, with additional input from other authors as required. We have made these details more explicit with the following highlighted amendments.

1. P 9 para 3 – added an additional final sentence as follows:

Although every effort was made to identify participants who were not known to the principal researcher, being someone who has worked in the medical billing industry for over 30 years it was likely that some participants would have a coexisting relationship. One GP and one SMO were personally known to the principal researcher, and another GP and SMO were professionally known. In addition, three SMOs were professional acquaintances. While this was unavoidable, it is not uncommon in qualitative research projects (for example a nurse questioning other nurses in their organisation as part of a project). 

1. P 9, we have added a new first paragraph and made a minor change to the first sentence of the second paragraph as follows:

To ensure personal relationships (none of which were close) did not cloud data collection, the first author continued to have regular discussions with other members of the research team adopting reflective practice to eliminate bias and ensure research integrity. Further, the third author listened to the audio recordings of all interviews and provided important insights when reviewing the draft paper to ensure data were accurately reflected and reported, with additional input from other authors as required.

To address any potential issues such as possible conscious or unconscious bias, triangulation was used where an experienced qualitative researcher separately analysed and interpreted the data and any differences in researcher perspectives were cross checked to arrive at an overall interpretation. By implementing these accepted methods rigour, trustworthiness, authenticity and credibility were addressed.

2. P 9, last para, we have made the following highlighted additions:

As this study forms part of the doctoral thesis of the principal researcher, it was incumbent upon her to personally conduct as much of the work as possible. However, this project was at all times closely supervised by the last author, who is a senior researcher experienced in qualitative data collection. The principal researcher had ongoing discussions with the last author throughout the data collection phase and during the analysis and coding of the data. 

3. P 10, first para – we have made the following highlighted changes and additions:

Further, to ensure research integrity the last author directly sat in and supervised the first two interviews (including with the GP who had a personal relationship). were directly supervised by the last author, who is a senior researcher experienced in qualitative data collection. Following approval of the first two interviews, the principal researcher and first author continued and personally conducted all 27 interviews. Most of the interviews were conducted in person (n = 23) at a place and time convenient to the participants. Due to geographical barriers, some of the regional GP interviews were conducted by phone (n = 4). 

Also, you were the only researcher. Having two researchers looking for themes and reaching agreement is a stronger methodology. 

We thank the reviewer for their comments and realize we were unclear on this point which led the reviewer into error. Whilst the first author was the principal researcher on this project, the first author was not the sole researcher. For clarity, this paper represents an important phase of the first author’s PhD project, and due to academic requirements, it was incumbent upon the first author to personally conduct as much of the work as possible. However, this project was at all times closely supervised by the principal PhD supervisor (last author) with whom the first author had ongoing discussions throughout the data collection phase and during the analysis and coding of the data. In addition, to ensure research integrity the principal supervisor directly sat in and supervised the first two interviews, which was stated in the paper. There were also two listeners and two independent coders, in line with qualitative research norms. The third author listened to the audio recordings of all interviews and edited final transcripts to ensure accuracy. After discussion with the principal supervisor regarding emergent themes, the first and third authors worked together to code the data, with the other authors reviewing in areas that required resolution of disagreement. We have made these details more explicit in the following highlighted amendments. 

4. P 10, we have inserted a new para 2 as follows:

Two listeners and two independent coders analysed the data in line with qualitative research norms. The third author listened to the audio recordings of all interviews and edited final transcripts to ensure accuracy. After discussion with the last author regarding emergent themes, the first and third authors worked together to code the data, with the other authors reviewing in areas that required resolution to disagreements. 

5. P 31, we have made the following highlighted addition:

However, any impact is likely minimal as all participants worked and billed independently day to day, and most did not know each other. Seven of the participants were known to the principal researcher either directly or indirectly, however, any impact is also likely minimal because the line of questioning was consistent across all participants and results were cross checked multiple times by multiple researchers using the recognised methods already discussed.

6. Medical schools by and large have no course work for clinicians in the healthcare systems within which they practice. I appreciate that you are calling attention to the problem, and presenting the experiences of the clinicians.

We thank the reviewer for their comments. We agree that these areas provide important and interesting avenues for further research and discussion and appreciate the insights of the reviewer.

REVIEWER 2:

The topic of administrative costs of health systems is of considerable interest. Australia is an important use case to understand health care systems from a comparative basis since the Medicare system is a national health system with interesting public and private components. This paper is a survey of practitioners in Australia inquiring about their knowledge of the billing process. It is well written, and of interest to people working in this field of administrative costs. However, despite its strengths, the manuscript is likely of limited interest to a broader medical audience. First, there is little explanation of the context of the Australia system for readers to understand Medicare and private. 

We thank the reviewer for their comments and insights. We concur that the issues highlighted in the article are of importance to cost administrators but suggest the interest extends well beyond that group of individuals, and is of great interest and importance to health policy-makers, health informaticians, health system managers, governments, health economists and health administrators because Universal Health Coverage systems are dependent on payment integrity and prevention of leakage. The WHO has stated this topic is of great importance, noting health system leakage has been described as ‘the last great unreduced heath care cost’. The global literature review of this topic forming part of the first author’s PhD identified a growing body of international literature, particularly in the U.S (referenced throughout this paper) on the importance of medical billing compliance in multiple jurisdictions and the serious gaps in education and support for medical practitioners in this area. We have made major changes throughout the manuscript to make these details more explicit with the following highlighted amendments. 

7. P 3, Abstract, Background, we have made the following changes to introduce a more international flavor to the paper:

Medical billing errors and fraud have been described as one of the last “great unreduced healthcare costs,” with some commentators suggesting measurable average losses from this phenomenon are 7% of total health expenditure. In Australia, it has been estimated that leakage from Medicare caused by non-compliant medical billing may be 10-15% of the scheme’s total cost. Despite a growing body of international research, mostly from the U.S, suggesting that rather than deliberately abusing the health financing systems they operate within, medical practitioners may be struggling to understand complex and highly interpretive medical billing rules, there is a lack of research in this area in Australia. The aim of this study was to address this research gap by examining the experiences of Australian medical practitioners through the first qualitative study undertaken in Australia, which may have relevance in multiple jurisdictions. as they interact with Australia’s Medicare by engaging in conversations with them about their lived experiences conducting medical billing in grass roots practice

8. P 4, Abstract, Conclusion, we have continued this line of discussion by making the following additions to the second and last sentences of the paragraph, as follows

The qualitative data presented in this study suggest Australian medical practitioners are ill-equipped to manage their Medicare compliance obligations, have low levels of legal literacy and desire education, clarity and certainty around complex billing standards and rules. Non-compliant medical billing under Australia’s Medicare scheme is a nuanced phenomenon that may be far more complex than previously thought and learnings from this study may offer important insights for other countries seeking solutions to the phenomenon of health system leakage. Strategies to address the barriers and deficiencies identified by participants in this study will require a multi-pronged approach. The data suggest that the current punitive system of ensuring compliance by Australian medical practitioners is not fit for purpose.

9. P 4, Introduction, we have added a new opening sentence with a reference to international research on this topic and deleted two complete sentences. This necessitated renumbering of all references, which we have also done:

Medical billing errors and fraud have been described as one of the last “great unreduced healthcare costs” with some commentators suggesting measurable average losses from this phenomenon are 7% of total health expenditure.1 Medical practitioners are often regarded as the custodians of health financing systems such as Australia’s Medicare because a significant portion of health budget distribution takes place at the transaction level pursuant to their myriad daily clinical decisions. It is therefore central to the long-term economic viability of any health system that medical practitioners have clarity and certainty around relevant billing standards and rules. However, a growing body of international research, mostly from the U.S, suggests medical practitioners are ill equipped to understand the complexities of the health systems in which they work. evidence suggests that medical billing rules in Australia are complex and medical practitioners may be experiencing difficulty navigating them.2

10. P 6 and 7 we have deleted most of the content completely and rewritten this section substantially, which we believe provides a more concise and international overview of the problem. 

Like the reported experiences of their U.S colleagues, evidence suggest Australian medical practitioners may be experiencing difficulty navigating complex medical billing rules.2 It has been suggested that the rate of non-compliant billing under Australia’s Medicare caused by deliberate abuses by medical practitioners is between 10-15%.3 However, how much non-compliant billing is deliberate is uncertain, as it rests in a spectrum with criminal fraud at one end and unintentional errors at the other and currently the precise quantum of each is unknown.4 This is largely because the problem is not what can be seen, but what cannot. Lax regulation, government maladministration, system complexity and the fact that medical practitioners are never taught how to use the system correctly at any point in their careers have all been cited as factors contributing to this problem.4 Increasing complexity has occurred in tangent with increased penalties for non-compliance5 and pressure on medical practitioners to bill correctly has reached the point where some authors have suggested that compliance with Medicare billing rules has become a contributing factor to medical practitioner burnout and suicide.6 However, one area of activity that has been overlooked is improving user knowledge of the medical billing system.

Multiple recent U.S studies on the topic of medical billing literacy7 have consistently reported demonstrably low literacy which may be improved by targeted educational initiatives, including by medical billing and coding education being a mandatory inclusion in the medical curriculum. However, an apparent inertia to act persists. In Australia, discussion around this topic is less mature, with very little similar research having been undertaken. 

The aim of this study was therefore to address this research gap by examining the experiences of Australian medical practitioners in grass roots practice as they interact with Medicare and claim reimbursements under Australia’s unique Medicare Benefits Schedule (MBS) codes.8

11. P 31, first para, we have expanded the first sentence to make clearer that the results are broadly consistent and comparable with experiences in the U.S and Canada, with references to important recent papers in those jurisdictions, as follows:

However, the qualitative data is contextually limited by the Australian context of a predominantly fee-for-service payment structure so the findings may not be generalisable, though the results are broadly comparable and consistent with reports of the same phenomenon in both the U.S and Canada.14-16 Another limitation is the potential impact of selection bias caused by the recruitment methods wherein a participant with high ethical standards was likely to work in a practice with others having the same standards. However, any impact would have been limited to the three GP practices where more than one GP was interviewed and possibly in the public hospitals where multiple SMOs were interviewed.

12. P 32, second para, we have made the following amendments to highlight the relevance of this topic to a broader international audience:

This is the first Australian study to examine the lived experiences of Australian medical practitioners interacting with Medicare and medical billing. Some of the experiences are shared with international experiences, and may therefore offer learnings for other countries implementing universal health coverage systems, in which payment integrity and control of system leakage are of critical importance. though moderated by Australia’s unique blended funding arrangements. The data suggest that the current system of ensuring compliance by medical practitioners in Australia is not fit for purpose.

Second, there is a significant amount of jargon here that is unique to the Australia context (gap, etc). 

We thank the reviewer for their comments and concur terms such as ‘gap’ and bulk billing’ are unique to the Australian market. We had already partly addressed this issue in the references. For example, reference 11 explains what ‘item 23’ is and reference 12 explains ‘bulk billing’, however we agree that more information will be helpful for readers. We have therefore made the following additional highlighted changes in the revised manuscript:

13. Reference 8 now describes Australia’s unique MBS billing codes which are unrelated to the ICD or CPT or any other code set and reference 12 now explains that a gap may be referred to as a co-payment or out of pocket patient cost in other countries. 

14. P 8, second para, we have added a phrase to explain that ‘SMO’ are specialists (not GPs), as follows:

Twenty-seven interviews were conducted, twelve with General Practitioners (GP) and fifteen with Salaried Medical Officers (SMO), the latter of whom are specialists working in Australian public hospitals. Participants were recruited through advertising with their professional associations, direct approaches and “snowballing”. Participant demographics included 11 females and 16 males and a mix of overseas and Australian trained medical practitioners, who worked in both regional and city locations. The full spectrum of career stages was represented, including early career stage medical practitioners (defined as 0-7 years post-graduation) through to those who had practiced medicine for over 30 years. The SMO cohort included a variety of procedural and non-procedural specialists. 

15. P 25, we deleted a large section of content where we concurred with the reviewer that it was too Australia specific and did not add to the paper. The deleted section is below.

The potential compliance impact of this phenomenon in the context of increasing corporatisation of the medical market, was in fact the trigger for the introduction of the Shared Debt Recovery Scheme already mentioned. However, the downstream effects of this reactionary approach by the Australian Government may be increased exposure to costly legal challenges similar to those that have already plagued the government’s Medicare compliance agency, the Professional Services Review (PSR), since inception,20 for alleged failures of due process. However, the next phase of legal challenges against the government as a result of this strategy are likely to be fought by larger corporate entities with both the time and liquidity to litigate and demand due legal process. This is in contrast to individual medical practitioners who may be unable to withstand the pressure and costs of litigation, and who may be forced to settle when their medical indemnity insurance organisations withdraw legal support.18 The first such corporate success was recently won by an after-hours medical service provider, who successfully argued it had been denied procedural fairness by the PSR.21 

We repeated this exercise and deleted another large Australia-centric section on pages 29 and 30 as follows.

A further recent example of a similar nature can be seen in the Australian Government’s manner of introducing new services to deal with the COVID-19 pandemic. The government used subordinate legislation for this purpose and sought to make bulk billing mandatory, when it is enshrined in the law as voluntary.1 This created an unprecedented and troubling inconsistency between the new subordinate COVID Determinations and the Australian Constitution as well as the Health Insurance Act 1973 (Cwth)27 and was unhelpful at a time when doctors were under extreme clinical pressure. By adding to medical practitioner confusion about what was or wasn’t compliant billing of the new COVID services, they were left second guessing risks relating to unknown but potentially serious downstream penalties for noncompliance.28

 Finally, it is clear that the findings are an important issue for policy makers in Australia to consider, but it’s not clear that there are generalizable lessons for other markets

We thank the reviewer for their comments and feel we have addressed this concern in points 7-15.

---

## [Decision Letter · Decision Letter 1]

13 Sep 2021

PONE-D-20-16346R1

Wading through Molasses: A qualitative examination of the experiences, perceptions, attitudes, and knowledge of Australian medical practitioners regarding medical billing

PLOS ONE

Dear Dr. Faux,

Thank you for submitting your manuscript to PLOS ONE. After careful consideration, we feel that it has merit but does not fully meet PLOS ONE’s publication criteria as it currently stands. Therefore, we invite you to submit a revised version of the manuscript that addresses the points raised during the review process.

We sincerely apologize for the delayed peer-review process. We have not been able to promptly find available Academic Editors and peer-reviewers to handle this Appeal process. However, one of the previous peer-reviewers agreed to re-review your manuscript and they are satisfied that all previous comments have been adequately addressed. Their comments are included below. They recommended that certain remaining jargon, such as bulk billing, are directly defined. They also suggest that you add a table highlighting the significant themes emerging from the interviews.

Upon internal review of your manuscript, we identified two additional points that we require you to address:

1) The paragraph ‘Government maladministration’ includes a rather strong criticism towards the Australian government. We recommend that you consider toning down the language of this paragraph, ensuring that all statements made are adequately related to and supported by the data showed in the manuscript. For example, the statement “It would appear the Australian Government is either unable or unwilling to explain the very medical billing laws it promulgates […]” appears to be too speculative and may attract undue external criticism.

2) Please include additional information regarding the interview guides used in the study and ensure that you have provided sufficient details that others could replicate the analyses. For instance, if you developed an interview guide as part of this study and it is not under a copyright more restrictive than CC-BY, please include a copy, in both the original language and English, as Supporting Information.

We look forward to receiving your revised manuscript.

Kind regards,

Dario Ummarino, Ph.D.

Senior Editor

PLOS ONE

Journal Requirements:

Reviewers' comments:

Reviewer's Responses to Questions

**Comments to the Author**

1. If the authors have adequately addressed your comments raised in a previous round of review and you feel that this manuscript is now acceptable for publication, you may indicate that here to bypass the “Comments to the Author” section, enter your conflict of interest statement in the “Confidential to Editor” section, and submit your "Accept" recommendation.

Reviewer #2: All comments have been addressed

2. Is the manuscript technically sound, and do the data support the conclusions?

Reviewer #2: Yes

3. Has the statistical analysis been performed appropriately and rigorously? 

Reviewer #2: N/A

4. Have the authors made all data underlying the findings in their manuscript fully available?

Reviewer #2: Yes

5. Is the manuscript presented in an intelligible fashion and written in standard English?

Reviewer #2: Yes

6. Review Comments to the Author

Reviewer #2: Thank you for your response to the initial reviews. They manuscript is much improved as a result.

There is still a significant amount of jargon in this revised manuscript. Please consider defining all of the terms (such as bulk billing and gap), or add a glossary of terms for the reader (or both). These terms are not in common use outside of Australia.

You may want to consider a table highlighting your significant themes for the reader.

7. PLOS authors have the option to publish the peer review history of their article (what does this mean?). If published, this will include your full peer review and any attached files.

Reviewer #2: No

---

## [Author Response · Author response to Decision Letter 1]

25 Sep 2021

Dear Editors,

RE: PONE-D-20-16346R1

Wading through Molasses: A qualitative examination of the experiences, perceptions, attitudes, and knowledge of Australian medical practitioners regarding medical billing.

Thank you for the opportunity to revise the above paper. We have addressed each of the reviewer comments below. We have also attached the revised paper with tracked changes.

We trust that we have satisfactorily addressed the minor revisions requested, and look forward to hearing from you shortly.

Yours sincerely

Margaret Faux (on behalf of all authors)

 

REVIEWER #2:

1) There is still a significant amount of jargon in this revised manuscript. Please consider defining all of the terms (such as bulk billing and gap), or add a glossary of terms for the reader (or both). These terms are not in common use outside of Australia.

We thank the reviewer for their comments and have made significant changes to address this issue. We have not added a glossary because by removing a considerable amount of Australia specific jargon (described below), there were only two words requiring definition (bulk billing and gap), both of which we have explained and addressed with the following highlighted amendments.

1. P 11 para 2 – added additional bracketed text as follows:

“…when I did my GP training we had a block of training prior to our very first day on the job…we basically just learnt you know your 23 and 36 item number [common time based attendances]11... there would have been question and answer time, but we hadn’t practised yet so we wouldn’t really have known what questions to ask.” (GP1) 

2 P 12 para 4, we have added the following clear explanation of bulk billing (while also retaining further explanation in the reference) as well as providing a simple explanation of a gap, as follows:

When participants were asked detailed questions about fundamental legal requirements to bill correctly, their levels of literacy were variable and some were confused in important areas, such as when it is permissible to charge a gap and what bulk billing was. Bulk billing is a common term in Australia, describing a transaction for a medical service wherein the patient does not pay any money because the medical practitioner chooses to accept the amount of the available government subsidy for that service.12 The term ‘gap’ in the Australian context refers to a patient out-of-pocket payment which in many countries is described as a co-payment. 

3 P 14 para 2, we have made the following highlighted additions to explain another element of Australian medical billing more explicitly:

The majority of participants were also unclear about fundamental billing requirements. In Australia’s gatekeeper model health system, patients usually require a valid referral from a general practitioner before seeking more specialised care. However, most participants did not know what constituted a valid referral. Other very basic requirements to bill correctly were also poorly understood by most participants such as specific rules around billing eligible war veterans, and whether any patient has to sign a form when the medical practitioner bulk bills the patient. 

4 P 23, last para – we have removed the following Australia specific highlighted text:

The data also suggest the current ‘rules’ of medical billing are confusing, and medical practitioners are struggling to understand and apply them in daily practice. Available evidence also suggests that recent Australian Government initiatives such as the MBS Review Taskforce (MBSRT)18 may be exacerbating these problems by making it difficult for medical practitioners to keep pace with the Australian Governments’ frenetic law making.2 

5. Page 25 last para and page 26 first two paras – we have deleted the following Australia specific content completely:

Another potential impact on public hospitals caused by disparate Federal and State agencies, is that well-intentioned, but siloed initiatives such as the MBSRT, may cause disruptions to legitimate revenue streams for State Governments when clinical code sets diverge. In a recent example, changes to the MBS colonoscopy items recommended by the MBSRT were introduced on 1 November 2019. These changes turned one MBS item number into seven, none of which matched the Australian Classification of Health Intervention Codes (ACHI).21 ACHI codes have a number of purposes including to code private patient encounters in public hospitals and when the codes submitted by a medical practitioner using the MBS do not match the codes submitted by the hospital using ACHI, for the same episode of care, Private Health Insurers may reject or delay payment, incorrectly assuming either the hospital or the medical practitioner has submitted a non-compliant bill. 

ACHI codes are updated biennially and despite being derived from the MBS, often differ from the MBS for the very same service, because ACHI represent different concepts intended for different use cases and are the responsibility of the Independent Hospitals Pricing Authority (IHPA).21 IHPA is focussed on hospital morbidity and mortality, whereas members of the MBSRT were predominantly medical practitioners, who were understandably focussed on writing MBS service descriptions that meant something to them. In addition to impacting State Government revenue streams, this increasing divergence in our national clinical classifications and code sets may ultimately hamper full implementation of Australia’s National Digital Health Strategy, which envisions standard semantic interoperability.22 To prevent this, it will be critical to ensure future code committees include individuals with the necessary skills to understand e-enabled health environments and work collaboratively aligning their codes with each other and with additional international codes already in use in Australia, such as SNOMED-CT and ICD-10AM.

6. Page 29, second para, we have removed the below highlighted text to align the changes described above.

Non-compliant medical billing under Australia’s Medicare scheme is a nuanced phenomenon that may be far more complex than previously thought. Therefore, many of the current punitive, post payment audit initiatives of the government are unlikely to succeed. such as trying to nudge medical practitioners into compliance with non-existent or incomprehensible rules they have never been taught and do not understand. 

7. Page 29, last para – we have made the below highlighted changes to align the changes described above.

Strategies to address the barriers and deficiencies identified by participants in this study will require a multi-pronged approach which may include the development of clear, legally binding medical billing rules, nationally consistent, accurate and accessible education, and structural reform to tightening and alignment of the underlying regulatory framework. including aligning national code sets.

8. Page 30 second para, we have removed the acknowledgment of Heather Grain because we have removed all reference to the Australia specific areas which we discussed with her.

The authors acknowledge the contribution of Heather Grain: Health Informatician, Clinical Coder, Health Information Manager, Digital Health Expert, for her input into Australia’s digital health environment and ACHI codes. The authors also wish to thank the medical practitioners who shared their experiences with medical billing and Medicare. The project received no funding.

2) You may want to consider a table highlighting your significant themes for the reader.

We thank the reviewer for this suggestion and have added a table (Table 1), which occupies the whole of page 10, displaying raw data and its analysis to themes. The addition of the table also necessitated moving two large quotes from the body of the paper and placing them in the table. These are highlighted in the table, which is copied in full below: 

Table 1

Raw Data Theme

[Interviewer asked SMO7 if education at various levels adequately equipped him to bill correctly] Not at all. It is purely through by necessity to understand it oneself and to understand the vagaries not only of billing, but how it works in the context of the staff specialist or ward arrangements, which are quite complex. [interviewer: ‘any education on that either?’] No zero. Zip. Inadequate induction 

Bulk billing, I understand is where whatever Medicare says, so if … I treat the patient for say keeping on breathing machine let us say. Government says you can earn $50 a day for doing that and bulk bill would be if I say okay give me $50. If I charge $60, then I have charged a gap. [interviewer: when can you do that? SMO12 replied] No idea. Poor legal literacy 

[interviewer]…so when it goes off into accounts, how confident are you about what happens next? [SMO14] I am confident because as the director, I have explored that, my colleagues would be somewhat less confident. [interviewer] With item numbers…? [SMO14] No just total numbers. Just money. Could have been anything. So, in fact, in reality I have no idea. [interviewer] So…you have got an idea of the total dollar amount that is billed, do you have an idea of the actual item numbers? [SMO14] No, not at all, not a jot, not one single solitary scintilla. Absence of reliable advice and support

[GP3] We have a practice manager and we have asked her to contact Medicare about some…uncertain issues regarding Medicare…and she will get five different answers from five different people that she rings…that is a regular experience and I say “…there’s no point in ringing Medicare about this” because I do not know who she is speaking to. I do not know whether she is speaking to a manager…or somebody who has recently started in Medicare who does not have much experience…and is just reading from one part of the manual but doesn’t know the other parts…we’ve always had that experience if you ring up…the most recent example…charging through Medicare for overseas travel…she has spoken to several different people and received different answers from each one. 

[GP2] I probably underbill…I’m just going to do what I know is safe. Fear and Deference

[GP4] The threat of audit kind of hangs over… 

[SMO7] I do not order a lot of blood tests. I do not order a lot of scans. I am very interested in…evidence base, I am interested in doing what is needed, I try not to pander to anxiety, it’s very difficult, it is much easier to give in and just order a million tests…It is an impost on the national health, so I think there is a responsibility. 

[GP8] Sending some more resources …for educating the doctors, by various means be it sending them letters like case examples, emails, having some conferences around, you know, correct Medicare billing etc and educating doctors the implications of incorrect charging particularly over-servicing and fraud, I think that is very important. Doctors just learn from their colleagues and others, you know, we are hearing stories, it is not something they are actively involved in, so there should be an education process and may be even attaching some category points…if the doctors understood Medicare and I think that is very important. The system is there but is not enough education about it. Unmet opportunities for improvement

 

INTERNAL REVIEW:

1) The paragraph ‘Government maladministration’ includes a rather strong criticism towards the Australian government. We recommend that you consider toning down the language of this paragraph, ensuring that all statements made are adequately related to and supported by the data showed in the manuscript. For example, the statement “It would appear the Australian Government is either unable or unwilling to explain the very medical billing laws it promulgates […]” appears to be too speculative and may attract undue external criticism.

We thank the reviewer for their comments and have made the following highlighted amendments and deletions throughout the manuscript, to soften the language as suggested. We did not mean to suggest that maladministration was intentional or deliberate, but that factors including poor resourcing and support were contributing to the problem.

1. P 5 para 1, we have made the following changes:

This is largely because the problem is not what can be seen, but what cannot. Lax regulation, poor government maladministration, system complexity and the fact that…

2. P 19 para 2, we have made the following changes:

The participant was subjected to what appears to have been a mishandled audit by Medicare, who did not appeared to have misunderstood the operation of the rule, which at all relevant times was clearly described in the MBS. As a result of the audit and Medicare’s failure to explain to the SMO what she did wrong (which appears to may have in fact been nothing), the SMO changed her billing behaviour and is now billing incorrectly and costing Australian taxpayers more.

3. Page 27, the Title ‘Government maladministration’ has been replaced by ‘Inadequate government support’

4. Page 27 last para and page 28 first two paras, we have made the following changes:

Participants reported that It was apparent that the “the blind leading the blind” method by which medical billing information is disseminated may be perpetuating errors and myths. Further, the consistency in the experiences of the wide cross section of participants in this study supports a finding that extremely low levels of legal literacy in relation to medical billing is fact rather than hyperbole, and there may be creating a vortex of misinformation contributing to health system leakage.

Further, the data suggest that apparent maladministration a lack of administrative resources and support provided by the Australian Government appears to may have left medical practitioners with no place to go for legally accurate, reliable advice, meaning that despite due diligence, a medical practitioner may still fall foul of the law. In one case, a participant who described correct billing practices, appears to have been led had been billing correctly, was effectively led into incorrect billing by the Australian Government who may not have the appropriate resources to provide accurate interpretations of appeared to have understood its own rules to practitioners. 

It would appear the Australian Government is either unable or unwilling to explain the very medical billing laws it promulgates, and as such, courts and other authorities must give due consideration to the veracity of any submission made by a medical practitioner under investigation for incorrect billing, relating to ignorance of relevant requirements or the potential impact of third parties on their billing. The participants of this study were clear that expecting medical practitioners to comply with complex and mercurial billing laws without relevant skills or training was unrealistic. Moreover, it is suggested that denying medical practitioners access to clear, reliable advice and support training prior to imposing sometimes very serious sanctions is indefensible and may be inconsistent with common law principles of natural justice.24 

2) Please include additional information regarding the interview guides used in the study and ensure that you have provided sufficient details that others could replicate the analyses. For instance, if you developed an interview guide as part of this study and it is not under a copyright more restrictive than CC-BY, please include a copy, in both the original language and English, as Supporting Information.

We thank the reviewer for this suggestion. 

On page 8, last para, we have added the below text and have attached the question guide to our submission as Appendix 1:

The interviews were semi-structured, with a question sheet used to loosely guide questioning. A copy of the question guide is shown as Appendix 1.

In addition to all of the above changes, we have made minor grammatical changes which are shown on the tracked version of the article, and have adjusted the reference list to align the changes made. This has reduced the number of references to 21.

---

## [Decision Letter · Decision Letter 2]

20 Dec 2021

Wading through Molasses: A qualitative examination of the experiences, perceptions, attitudes, and knowledge of Australian medical practitioners regarding medical billing

PONE-D-20-16346R2

Dear Dr. Faux,

We’re pleased to inform you that your manuscript has been judged scientifically suitable for publication and will be formally accepted for publication once it meets all outstanding technical requirements.

Kind regards,

Kathleen Finlayson

Academic Editor

PLOS ONE

Additional Editor Comments (optional):

Reviewers' comments:

Reviewer's Responses to Questions

**Comments to the Author**

1. If the authors have adequately addressed your comments raised in a previous round of review and you feel that this manuscript is now acceptable for publication, you may indicate that here to bypass the “Comments to the Author” section, enter your conflict of interest statement in the “Confidential to Editor” section, and submit your "Accept" recommendation.

Reviewer #3: All comments have been addressed

2. Is the manuscript technically sound, and do the data support the conclusions?

Reviewer #3: Yes

3. Has the statistical analysis been performed appropriately and rigorously? 

Reviewer #3: Yes

4. Have the authors made all data underlying the findings in their manuscript fully available?

Reviewer #3: Yes

5. Is the manuscript presented in an intelligible fashion and written in standard English?

Reviewer #3: Yes

6. Review Comments to the Author

Reviewer #3: Extremely relevant study. Great presentation of data. It would be interesting to compare if there’s a regional variation amongst providers.

7. PLOS authors have the option to publish the peer review history of their article (what does this mean?). If published, this will include your full peer review and any attached files.

Reviewer #3: **Yes: **Anshul Arora, MD

---

## [Editor Report · Acceptance letter]

3 Jan 2022

PONE-D-20-16346R2 

Wading through Molasses: A qualitative examination of the experiences, perceptions, attitudes, and knowledge of Australian medical practitioners regarding medical billing 

Dear Dr. Faux:

I'm pleased to inform you that your manuscript has been deemed suitable for publication in PLOS ONE. Congratulations! Your manuscript is now with our production department. 

Kind regards, 

on behalf of

Dr. PLOS Manuscript Reassignment 

Staff Editor

PLOS ONE